# Bias in the perceived prevalence of open defecation: Evidence from Bihar, India

**Jinyi Kuang**[1]*, **Erik Thulin**[2], **Sania Ashraf**[1], **Alex Shpenev**[1], **Upasak Das**[3], **Maryann G. Delea**[4], **Peter McNally**[1], **Cristina Bicchieri**[1]*

**1** Center for Social Norms and Behavior Dynamics, University of Pennsylvania, Philadelphia, PA, United States of America, **2** Center for Behavior & the Environment, Rare, Arlington, VA, United States of America, **3** Global Development Institute, University of Manchester, Manchester, United Kingdom, **4** Gangarosa Department of Environmental Health & Hubert Department of Global Health, Rollins School of Public Health, Emory University, Atlanta, GA, United States of America

* jkuang@sas.upenn.edu (JK); cbicchieri@sas.upenn.edu (CB)

**Data Availability Statement:** All data and analysis files are available at the Open Science Framework (osf.io/f82tk/).

**Funding:** This project was funded by the Bill and Melinda Gates Foundation (Grant No. INV-009118 /

## Abstract

People often form perceptions about how prevalent a behavior is in a social group. However, these perceptions can be inaccurate and biased. While persistent undesirable practices in low-income countries have drawn global attention, evidence regarding people's perception of how prevalent these practices are is scarce. Among those harmful practices, open defecation in India remains a significant public health concern, where it perpetuates the vicious cycle of disease and poverty. In this study, we focus on measuring the perceived prevalence of open defecation among respondents in Bihar, India. We examined the bias in perceived prevalence, which is defined as a pattern of deviation from the actual prevalence of open defecation. Results showed that respondents who defecate in the open overestimate the prevalence of open defecation, whereas those who consistently use toilets underestimate it. This finding suggests a false consensus bias in the perceived prevalence of open defecation. Scholars, policymakers, and program implementers who seek to correct misperceptions about open defecation by broadcasting real prevalence should be aware of biases in the perceived prevalence and address them in behavior change interventions.

## Introduction

As social beings, humans often form perceptions about the prevalence of a particular behavior among members of their social groups [1, 2]. An individual's prevalence perception is highly subjective and may not accurately reflect the prevalence of certain socially undesirable behaviors [3–5]. Specifically, individuals' perceived prevalence regarding these behaviors can systematically deviate from their actual prevalence, which is referred to as the bias in individuals' prevalence perception [6]. It is well accepted that correcting the misperception about the prevalence of undesirable behaviors could be effective in motivating behavior change. However, the literature is less consistent in whether and how people might systematically *bias* their prevalence perception from the actual prevalence across behavior domains. Previous field studies have reported that people have the overall tendency to overestimate the prevalence of socially

OPP1157257). The funder was not involved in study design, data collection, data analysis, and data interpretation. Kantar Republic managed sample selection and data collection, but had no role in study design, decision to publish, or preparation of the manuscript.

**Competing interests:** Kantar Republic managed sample selection and data collection for this study. This does not alter our adherence to PLOS ONE policies on sharing data and materials.

undesirable behaviors, irrespective of their own behavior [7–9]. Conversely, another line of literature suggests that the bias in prevalence perception is related to people's own behavior. Specifically, people tend to show a false consensus bias, that is, perceiving that their own behavior is more common than it actually is [10, 11]. This means those who engage in an undesirable behavior could *overestimate* the prevalence of that undesirable behavior, whereas those who practice the desirable behavior could *underestimate* the prevalence of that undesirable behavior. These two different types of biases could have different policy implications for effectively designing behavior change programs that seek to correct the misperceptions. Therefore, it is important to assess whether and how people show bias in the perceived prevalence of the targeted behavior to mitigate the risk of intervention backfire.

In this paper, we focused on measuring and examining the perceived prevalence of defecation practices in India. While persistent undesirable practices in low-income countries have drawn global attention, evidence regarding people's perception of how prevalent these practices are is scarce. Among those harmful practices, open defecation in India remains a significant public health concern, where it perpetuates the vicious cycle of disease and poverty [12]. Recent field studies in India suggest that social factors, including social beliefs regarding others' sanitation behaviors, are related to one's own sanitation behaviors [13]. Such insights open avenues for designing interventions that leverage social influence. Therefore, understanding how an individual's perceptions regarding open defecation prevalence deviate from the actual prevalence of open defecation is crucial for behavior change programs seeking to curb open defecation by changing social beliefs. Specifically, if only those who defecate in the open overestimate the prevalence of open defecation whereas those who use a toilet underestimate it, broadcasting the actual prevalence of open defecation could potentially discourage individuals who defecate in a toilet, which may lead them to discontinue their toilet use [14]. In this study, we aimed to assess people's prevalence perceptions about open defecation and examine whether biases in the perceived prevalence of this behavior are associated with their reported defecation practices. We also discuss the implications of our findings for the design and targeting of interventions that seek to address social beliefs and encourage sanitation-related behavior change.

## Materials and methods

### Study context

In India, the prevalence and persistence of open defecation remain a long-standing issue with a known impact on public health. Poor sanitation has been linked to transmission of enteric diseases, as well as an exacerbation of stunting [15]. We conducted this study in Bihar, one of India's most populated states that continues to lag behind India's average in terms of toilet coverage and usage [16, 17]. We drew the data for our analyses from the Longitudinal Evaluation of Networks and Norms Study (LENNS) conducted in 2017 to 2018 in Bihar, India. This formative research aimed to elucidate social determinants related to sanitation behaviors [18, 19].

### Data collection

From April to June 2018, trained field workers administered a cross-sectional survey among respondents aged 16 to 65 years in thirty sampling units in Bihar, India. The study sample was drawn from three types of geographic regions, including six rural communities (Gram Panchayats), eighteen semi-urban communities (census wards from six Nagar Panchayats), and six urban communities (registered slums). A complete listing of dwelling units/households in the selected areas was generated, and respondents were randomly selected from each sampling unit. In this study, we use sampling units as proxies for communities: GP from rural; slum for

urban and census wards for semi-urban region. Further details regarding the sampling strategy have been published elsewhere [19].

All survey items were translated from English to Hindi and back-translated by a third party. Field workers fluent in the local language (Hindi) received training on standardized data collection procedures. All inconsistencies were addressed and revisions were made following a pilot study to ensure the survey questions reflected the study aim. All participants provided informed consent prior to data collection. The data were collected by Computer Assisted Personal Interviewing (CAPI) on hand-held tablets. The study was approved by the University of Pennsylvania Institutional Review Board (Protocol #:827239).

## Measurement

**Community-level measures.**  To assess the prevalence of community-level open defecation, trained field workers first asked about the respondent's last toilet use behavior: "Some people defecate in the open; some people use a toilet, where did you defecate the last time you had to?" The answers were coded as binary such that 0 represented respondents who used of any type of toilets and 1 represented respondents who defecated in the open. For each sampling unit, we averaged the answers across respondents as a proxy for community-level prevalence of open defecation.

**Individual-level measures.**  To measure individuals' perceived prevalence of open defecation in their community, participants were asked: "Out of ten members in your community, how many do you think defecated in the open the last time they needed to defecate?" The response was captured between 0 (low) and 10 (high) to reflect the levels of perceived open defecation prevalence. The survey framing was informed by cognitive interviews during the pilot. Field workers qualitatively tested different survey items framing to assess numeric comprehension and "Out of ten people in your community" was relatively easily understood for the perceived prevalence measure (See S1 File for further details). To assess respondent's comprehension of the survey questions, we also measured respondents' perceived prevalence of toilet use behavior with a similar style of question, asking "Out of ten members of your community, how many do you think used a latrine the last time they needed to defecate?" We then reverse-coded their responses by subtracting their answer from 10, such that the scale reflected respondents' perceived prevalence of people not using a toilet in their community. We excluded responses from individuals with a high level of discrepancy between these two measures, due to inadequate comprehension (See S1 File).

To assess the primary defecation practice among respondents, we asked them: "In the past week, how often have you used a latrine to defecate? Never, occasionally, frequently, or every time?" We categorized respondents into three groups, i) open defecation (never use a toilet in the past week), ii) inconsistent toilet use (occasionally or frequently used a toilet in the past week), and iii) consistent toilet use (used a toilet to defecate every time in the past week). In addition to these data, field workers also collected the respondents' demographic information including their gender, age, educational attainment, religion, caste groups they identified with, and household asset ownership. We used a number of household assets (color TV, fridge, motorcycle) owned by the household as a proxy of respondents' socio-economic status [15], ranging from extremely low socio-economic status (SES) (not owning any of the measured assets) to high SES (owning all three types of assets).

## Statistical analysis

We quantified biases regarding perceptions of open defecation prevalence, which is defined as the discrepancy between individual-level prevalence perception and community-level

prevalence of open defecation. To do so, we first linearly transformed the individual-level prevalence perception measure from 0–10 to 0–1 to keep it on the same scale as a community-level measure. Then we subtracted it from the community-level open defecation prevalence. The bias in the perceived prevalence ranged from -1 to 1, where negative scores represented underestimation of open defecation prevalence, positive scores represented overestimation of open defecation prevalence, and 0 represented accurate perception of open defecation prevalence. The computational procedure was reflected in Fig 1. To quantify how accurate an individual's prevalence perception is, we took the absolute value of individuals' bias regarding the open defecation prevalence. To test the hypothesis that individuals' bias in the perceived open defecation prevalence is associated with their own defecation practice, we constructed a generalized linear model (GLM) with prevalence perception bias as the outcome variable and individual's defecation practice as the explanatory variable. We included gender, education, age, socio-religious group, and socio-economic status as covariates and adjusting for the sampling unit where respondents reside (See F1 file). All statistical tests were obtained at a significant level of 0.95. Statistical analyses were performed with R version 3.6 [20].

## Results

The field team surveyed a total of 2533 respondents. We excluded 120 respondents due to missing responses (n = 5) and evidence of inadequate comprehension of the norm perceptions questions (n = 115). The final analytical sample consisted of data from 2413 respondents (female = 53%) from 30 communities, representing an average of 80 respondents per community (SD = 44). The demographic characteristics, sanitation practices, and perceptions of open defecation practices in one's community of the study sample can be found in Table 1.

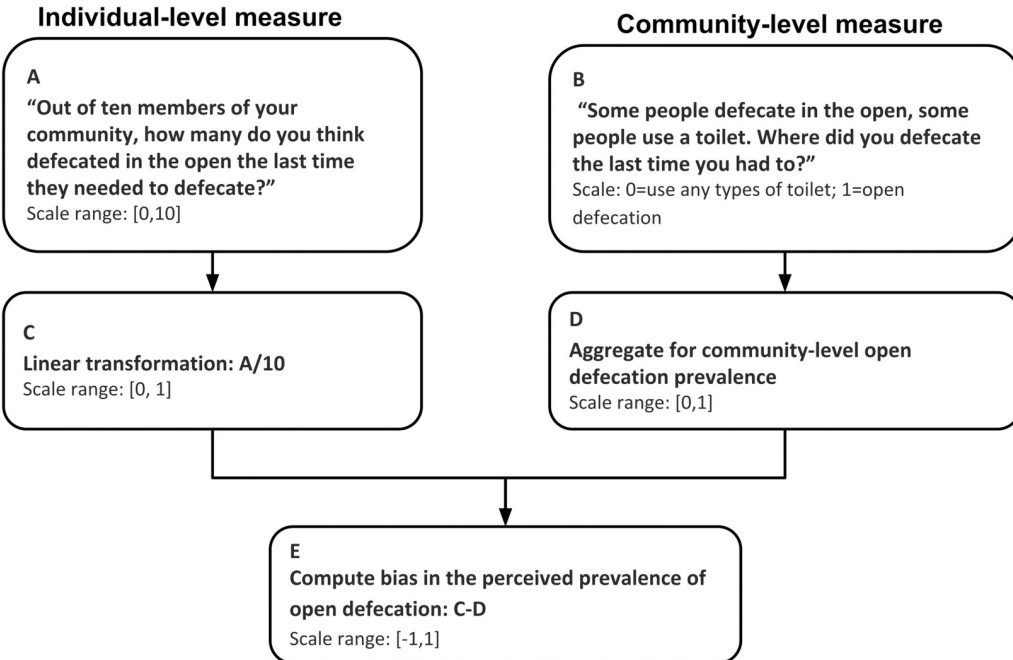

**Fig 1. Quantifying bias in the prevalence perception of open defecation behavior.** The prevalence perception bias score ranges from -1 to 1, such that a positive score indicates that respondents overestimate open defecation prevalence in their community (1 represent the largest possible overestimation bias), a negative score indicates underestimation (-1 represent the largest possible underestimation bias), and 0 represent accurate prevalence perception.

**Table 1. Demographic characteristics, sanitation practices, and perceptions of open defecation practices in one's community, of the study population, Bihar, India 2018.**

| | Bihar | Urban Slum | Semi-urban | Rural |
|---|---|---|---|---|
| | N = 2,413 | N = 812 | N = 811 | N = 790 |
| **Characteristics** | | | | |
| Age mean (sd) | 35 (14) | 34 (14) | 35 (14) | 36 (14) |
| Female n (%) | 1,272 (53) | 411 (51) | 417 (51) | 444 (56) |
| No formal education n (%) | 1,113 (46) | 368 (45) | 309 (38) | 436 (55) |
| **Socio-religious group n (%)** | | | | |
| Hindu-upper caste | 137 (6) | 45 (6) | 46 (6) | 46 (6) |
| Hindu-scheduled caste | 609 (25) | 402 (50) | 101 (12) | 106 (13) |
| Muslim | 612 (25) | 146 (18) | 160 (20) | 306 (39) |
| Others | 1,055 (44) | 219 (27) | 504 (62) | 332 (42) |
| **Social Economics Status (SES)[a] n (%)** | | | | |
| Extremely low SES | 1,202 (50) | 340 (42) | 359 (44) | 503 (64) |
| Low SES | 699 (29) | 298 (37) | 240 (30) | 161 (20) |
| Medium SES | 327 (14) | 114 (14) | 135 (17) | 78 (10) |
| High SES | 185 (7.7) | 60 (7.4) | 77 (9.5) | 48 (6.1) |
| **Last defecation event n (%)** | | | | |
| Used a toilet | 1,417 (59) | 545 (67) | 550 (68) | 322 (41) |
| Defecated in the open | 996 (41) | 267 (33) | 261 (32) | 468 (59) |
| **Last week's defecation practice n (%)** | | | | |
| Consistently toilet use | 1,163 (48) | 416 (51) | 475 (59) | 272 (34) |
| Inconsistently toilet use | 473 (20) | 209 (26) | 131 (16) | 133 (17) |
| Open defecation | 777 (32) | 187 (23) | 205 (25) | 385 (49) |
| **Perceived prevalence of open defecation mean (sd)** | 0.41 (0.36) | 0.34 (0.35) | 0.32 (0.31) | 0.59 (0.34) |
| Actual prevalence of open defecation across communities mean (sd) | 0.41 (0.3) | 0.33 (0.28) | 0.32 (0.25) | 0.59 (0.29) |
| Accuracy in the perceived prevalence of open defecation[b] mean (sd) | 0.17 (0.15) | 0.17 (0.15) | 0.17 (0.16) | 0.17 (0.13) |
| **Bias in the perceived prevalence of open defecation[c] n (%)** | -0.00 (0.23) | 0.00 (0.23) | -0.01 (0.23) | -0.01 (0.21) |
| Overestimation | 1093 (45.3) | 339 (41.7) | 338 (41.7) | 416 (52.7) |
| Underestimation | 1243 (51.5) | 473 (58.3) | 396 (48.8) | 374 (47.3%) |
| No bias (accurate) | 0 (0.0) | 0 (0.0) | 77 (9.5) | 0 (0.0) |

[a] We assessed SES using the number of household assets (color TV, fridge, motorcycle) owned by households. These three assets were used in the National Family Health Survey (NFHS), which is a nationally representative survey conducted throughout India by the Ministry of Family Health and Welfare, Government of India. We got similar results when using Principal Component Analysis (PCA) to estimate SES.

[b] Accuracy of the perceived prevalence of open defecation is defined as the absolute deviation of an individual's perceived community open defecation prevalence and the actual community open defecation prevalence based on self-reported data. A score of 0 indicates accurate perceptions of open defecation. Larger numbers indicate a higher level of misperception of open defecation prevalence.

[c] Bias in the perceived prevalence of open defecation is defined as the discrepancy of an individual's perceived community open defecation prevalence and the actual community open defecation prevalence based on self-reported data. Positive scores indicate an overestimation of actual open defecation prevalence, whereas negative scores indicate an underestimation of actual open defecation prevalence.

We found respondents show an overall inaccurate perception of open defecation prevalence in their communities (Table 1). Specifically, respondents' perception of open defecation prevalence deviated from the actual prevalence of open defecation by 17% on average (SD = 0.15). Regarding the direction of the misperception, we found 45.3% overestimated the OD prevalence and 51.5% underestimated the OD prevalence. In the multivariate analysis adjusting for the respondents' gender, age, educational attainment, socio-religion, and socio-economic status, we found that compared to those who reported defecating in the open in the week prior to

**Table 2. Association between bias in the perceived prevalence of open defecation and defecation practices, and individual characteristics, controlling for community-level fixed effects.**

**Bias in the perceived prevalence of open defecation**

| | Coef. (95% CI) |
|---|---|
| Defecation behavior (Ref. Open defecation) | |
| Inconsistent toilet use | -0.10***(-0.13, -0.08) |
| Consistent toilet use | -0.21***(-0.23, -0.19) |
| Gender (Ref. Women) | |
| Men | 0.02**(0.01, 0.04) |
| Education (Ref. No primary education) | |
| Primary | -0.03***(-0.05, -0.01) |
| Secondary | -0.03**(-0.06, -0.001) |
| High school | -0.05***(-0.09, -0.02) |
| College and above | -0.04**(-0.08, -0.004) |
| Age | -0.0005 (-0.001, 0.0002) |
| Socio-cultural group (Ref. Hindu upper caste) | |
| Hindu scheduled caste | 0.13***(0.09, 0.17) |
| Muslim | 0.10***(0.06, 0.15) |
| Others | 0.08***(0.04, 0.11) |
| Socioeconomic status (Ref. Extremely low SES) | |
| Low SES | -0.01 (-0.03, 0.01) |
| Medium SES | -0.04***(-0.06, -0.01) |
| High SES | -0.04**(-0.08, -0.01) |

N = 2,413; $R^2$ = 0.25; $R^2_{adj}$ = 0.23;

Residual Std. Error = 0.20 (df = 2369); F Statistic = 18.00***(df = 43; 2369); $p_{adj}<0.1^*$;$p_{adj}<0.05^{**}$;$p_{adj}<0.01^{***}$

survey administration, those who use a toilet inconsistently (b = -0.10, 95% CI: -0.13, -0.08, p<0.001) and those who use a toilet constantly (b = -0.21, 95% CI: -0.23, -0.19, p<0.001) have significantly lower estimation of open defecation prevalence than it actually is (Table 2). The predicted value showed that those who defecated in the open in the week prior to survey administration perceived that open defecation was 13% more prevalent than the actual prevalence within their communities. In contrast, those who reported consistent use of a toilet perceived that open defecation was 8% less prevalent than the actual prevalence in their communities (Fig 2).

## Discussion

In this study, we found evidence that individuals hold inaccurate perceptions of how prevalent open defecation is in their community, where some overestimate its prevalence and some underestimate its prevalence. We found that bias in the perceived prevalence of open defecation is associated with respondents' own defecation practices. Specifically, we found that respondents who defecated in the open tended to overestimate open defecation prevalence, whereas consistent toilet users tended to underestimate open defecation prevalence. These tendencies can be explained by the false consensus bias, where individuals overestimate the extent to which their beliefs, preferences, or behavior are more common among others [10, 11]. Our results highlight that the direction and levels of misperceptions of open defecation prevalence vary based on one's own behavior. This suggests that behavior change strategies that aim to correct misperceptions by broadcasting the actual prevalence of open defecation or toilet use to promote adoption of toilets may be subject to unintended consequences such as the

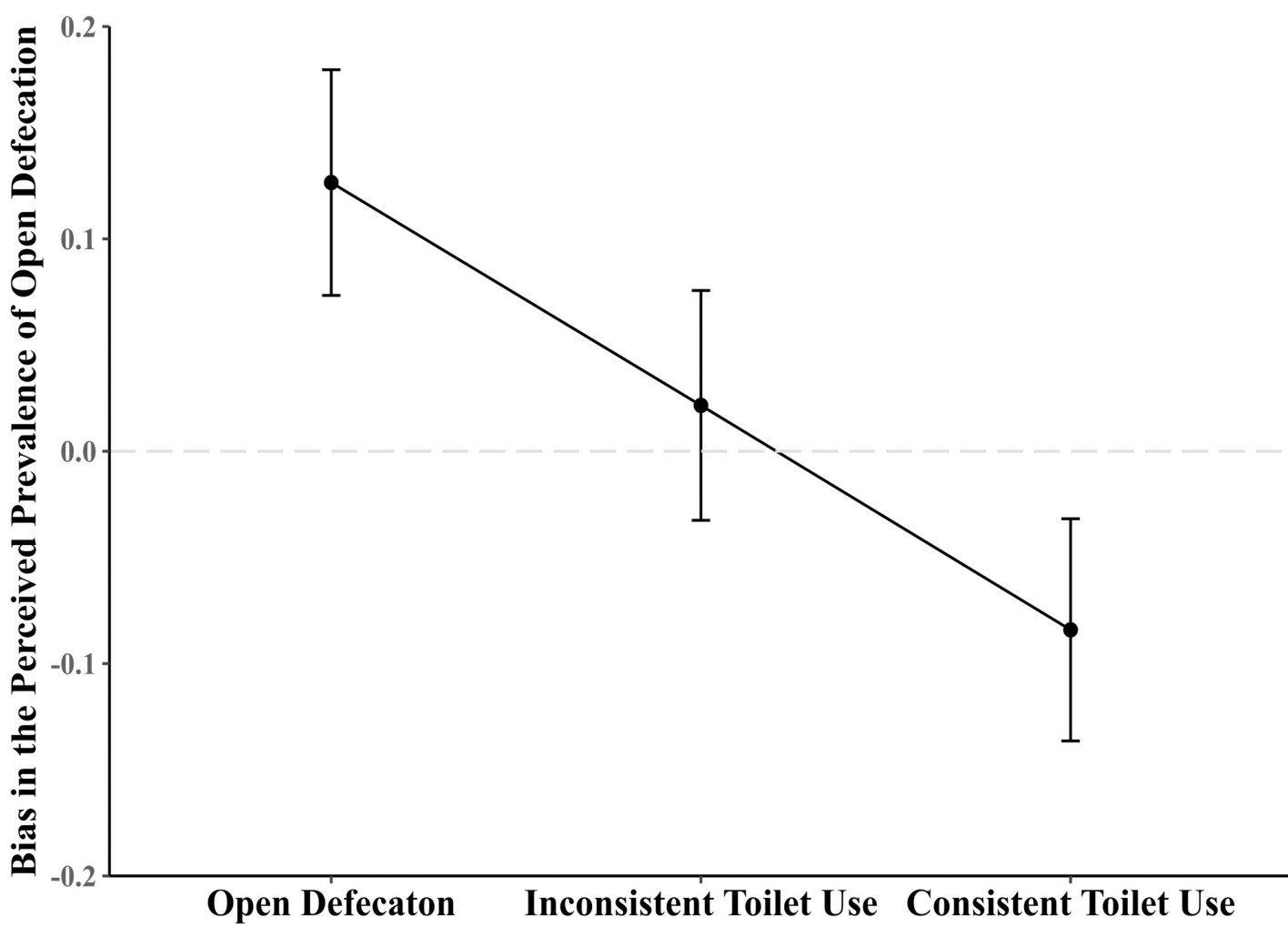

**Fig 2. Predicted value and confidence interval of the bias in the perceived prevalence of open defecation by individual's defecation patterns in the last week.** Bias in the perceived prevalence refers to the discrepancy (both direction and magnitude) of an individual's perceived community open defecation prevalence and the actual community open defecation prevalence proxied by self-reported data. Positive values indicate an overestimation, whereas negative values indicate an underestimation. This figure suggests that open defecators have the tendency to overestimate the prevalence of open defecation while consistent toilet users have the tendency to underestimate the prevalence of open defecation (Open Defecation M = 0.127, SE = 0.027; Inconsistent Toilet Use M = 0.022, SE = 0.028; Consistent Toilet Use M = -0.084, SE = 0.027).

boomerang effects. Boomerang effects were observed in previous studies of interventions encouraging energy conservation [14]. After receiving a perception correction message that describes the actual prevalence of the behavior (e.g., actual average energy use of their neighbors), those who already consumed a low level of energy (i.e., the desirable behavior) increased their energy use (i.e., adopted an undesirable behavior) [14]. To mitigate the risk of sanitation interventions backfiring (i.e., those who occasionally use a toilet stop using it after knowing that toilet use is less prevalence than they previously thought), we suggest that program implementers and researchers first examine the actual and perceived prevalence of the targeted behavior, and quantify an individual's prevalence perception bias (i.e., how one's perceived behavior prevalence deviates from the actual prevalence) using the method we outlined in this paper. Such an approach could help relevant stakeholders identify individuals who are more likely to hold biased perceptions regarding behavior prevalence, and correct misperceptions through the use of personalized interventions (e.g., household counseling, mobile phone reminders) [21].

Our study is not without limitations. First, we asked respondents about their beliefs regarding open defecation prevalence in their community. There may be subjective variation in estimating open defecation prevalence due to differences in the respondents' definition of what constitutes their 'community'. In cognitive interviews with community members, most said they understood the translated word for community to refer to those who lived in the area around where they live [18, 19]. However, future research is needed to understand the respondents' perception of social proximity, as community may mean spatially proximate others, but may also include groups with which one identifies who may not be spatially close. Second, the study site and India nationwide were subject to substantial sanitation promotion programs during the conduct of this study. Respondents may have under-reported open defecation practice due to social desirability bias. We tried to address this issue by asking balanced questions like "Some people defecate in the open; some people use a toilet, where did you defecate the last time you had to?" to reduce social desirability bias. We also collected observational data regarding toilet ownership and functionality, and cross-checked behavioral measures with these observational measures. Third, the cross-sectional design of this study does not allow for an examination of causation regarding whether behavior leads to norm perception bias or vice versa. On the one hand, it is possible that those who defecate in the open overestimate the prevalence of this behavior to *justify* their behavior. On the other hand, it is also possible that this bias encourages individuals to adjust their behavior to coincide with their subjective prevalence perception [22]. As reported in Bicchieri, et al., 2018, toilet use was strongly influenced by the belief that other members of one's reference network are using toilets [19]. Further research assessing this bias in the perceived prevalence should include mixed-method research techniques to verify findings in the sample population.

In conclusion, we provide field evidence that individuals hold inaccurate perceptions of open defecation prevalence in their community, and the bias in the prevalence perception is associated with their own sanitation practices. We also find individuals who self-identify as belonging to specific demographic groups differ in this bias in our study context. If social beliefs about what the majority of others do affect behavior, we should pay attention to how *biases in social beliefs* support or hinder behavior change. We suggest scholars, policymakers, and practitioners consider measuring the bias in the perceived behavior prevalence when designing behavior change interventions that leverage social influence.

## Supporting information

**S1 File. Supplemental details of measurements and analysis.**
(DOCX)

## Acknowledgments

The authors thank all study participants for volunteering their time to respond to the survey.

## Author Contributions

**Conceptualization:** Jinyi Kuang, Erik Thulin.

**Data curation:** Jinyi Kuang, Erik Thulin, Sania Ashraf, Alex Shpenev, Upasak Das, Peter McNally, Cristina Bicchieri.

**Formal analysis:** Jinyi Kuang.

**Funding acquisition:** Cristina Bicchieri.

**Investigation:** Jinyi Kuang, Erik Thulin, Sania Ashraf.

**Methodology:** Jinyi Kuang, Maryann G. Delea.

**Project administration:** Jinyi Kuang, Sania Ashraf, Alex Shpenev, Maryann G. Delea, Peter McNally.

**Resources:** Peter McNally, Cristina Bicchieri.

**Software:** Jinyi Kuang.

**Supervision:** Erik Thulin, Sania Ashraf, Alex Shpenev, Upasak Das, Maryann G. Delea, Peter McNally, Cristina Bicchieri.

**Validation:** Erik Thulin.

**Visualization:** Jinyi Kuang.

**Writing – original draft:** Jinyi Kuang.

**Writing – review & editing:** Jinyi Kuang, Erik Thulin, Sania Ashraf, Alex Shpenev, Upasak Das, Maryann G. Delea, Peter McNally, Cristina Bicchieri.

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
