## [Decision Letter · Decision Letter 0]

3 Aug 2020

PONE-D-20-16576

Bias in the Perceived Prevalence of Open Defecation: Evidence from Bihar, India

PLOS ONE

Dear Dr. Kuang,

Thank you for submitting your manuscript to PLOS ONE. After careful consideration, we feel that it has merit but does not fully meet PLOS ONE’s publication criteria as it currently stands. Therefore, we invite you to submit a revised version of the manuscript that addresses the points raised during the review process.

I enjoyed reading your paper, and I believe it touches an important topic. Reviewer #1 provides a few interesting comments (mainly related to making further clarification) which you may wish to address in your paper. I’m looking forward to an updated version of your manuscript.

We look forward to receiving your revised manuscript.

Kind regards,

Tomáš Želinský, Ph.D.

Academic Editor

PLOS ONE

Journal Requirements:

2. We note that informed consent was obtained from all participants of the research study. As we understand that participants as young as 16 year old were involved, please specify if consent from guardians was obtained in those instances

Additional Editor Comments (if provided):

Reviewers' comments:

Reviewer's Responses to Questions

**Comments to the Author**

1. Is the manuscript technically sound, and do the data support the conclusions?

Reviewer #1: Yes

Reviewer #2: Yes

2. Has the statistical analysis been performed appropriately and rigorously? 

Reviewer #1: Yes

Reviewer #2: Yes

3. Have the authors made all data underlying the findings in their manuscript fully available?

Reviewer #1: Yes

Reviewer #2: Yes

4. Is the manuscript presented in an intelligible fashion and written in standard English?

Reviewer #1: Yes

Reviewer #2: Yes

5. Review Comments to the Author

Reviewer #1: This paper examines individual’s perception about the prevalence of open defecation within their communities and studies whether it is misperceived. The paper studies the context of India, where open defecation is a significant sanitation issue. To examine this research question, the authors conduct a large survey eliciting individual prevalence of open defecation (a measure they use to create a proxy for actual defecation in the community) and perceptions about the rate of open defecation within the community. The authors show that individuals over or underestimate the community rate of open defecation based on their own actions. I like the paper, it studies an important topic but I feel at times the exposition is unclear and needs further clarification. I suggest some amendments below which should hopefully improve clarity.

Comments

How representative is the sample? This is important if the authors are trying to argue that their community level measure is a proxy for the community level norm. Can the authors compare characteristics against those in the census for example?

Direct response questions asking about individual open defecation may suffer from socially desirable response bias while those about the community are less likely to suffer from such a bias. Do the authors think this is an issue? If so a caveat should be added in their results or conclusion.

For their community level measure the authors use the questions “"Some people defecate in the open; some people use a toilet, where did you defecate the last time you had to?"” and then average across all respondents in the community. What is the distribution of these responses? Are the results robust to using the median? And importantly, are the results robust to using the community average of the question “In the past week, how often have you used a latrine to defecate? Never, occasionally, frequently, or every time?"” (as a community level measure). The reason I ask is that I am a little concerned with the time frame chosen by the authors for their community measure and the individual perceived prevalence. It is highly possible that people use both a toilet and open defecate and this intensity would not be accounted for with their current measures. To test if it matters, one can compare my suggested bias (community open deffication in the past week with perceptions about community defecation the last time) with the currently used bias.

The authors may want to cite the recent paper by Leonardo Bursztyn Alessandra L. Gonzalez and David Yanagizawa-Drott. Misperceived Social Norms: Female Labor Force Participation in Saudi Arabia (accepted at American Economic Review) which focuses on differences between individual beliefs and perceptions about the community.

In the introduction the authors only specifically mention false consensus bias in the discussion of the results. Since this paper is essentially about this bias, it may be useful to mention this theory earlier in the introduction.

I assume the footnotes for Table 1 are incorrect please check (there are two references to a footnote a and no reference to a footnote c). On this note, I am a little confused about the differences between Bias in the perceived prevalence of open defecation and the Accuracy of the prevalence perception of open defecation (Table 1). The authors need to discuss the differences between these two definitions and results. The authors may want to consider using different labels. Further, according to Table 1, the bias measure is zero suggesting (as the authors state on line 133 of page 7) individuals have an accurate perception of open defecation prevalence. This is quite striking and not particularly discussed in the results. And why do the authors focus on the accuracy measure when discussing Table 1 (ignoring discussion of the bias measure) and then fail to use this accuracy measure in their estimation (Table 2, Figure 2) instead turning back to the bias measure. This seems to be inconsistent and not well explained.

The authors should discuss results in Table 2 as they are important. At the moment the authors simply state “The analysis results were shown in Table 2.”

Can the authors include the mean for “actual community open defecation prevalence” in Table 1.

Where do Figure 2 results come from? Results appear to be different from those in Table 2

Reviewer #2: Interesting paper showing the false consensus bias regarding the prevalence of open defecation in Bihar, India. A useful starting point for further studies aiming to correct the misperceptions.

One minor comment: Superscripts in Table 1 referring to the Table notes are wrong (a,a,b instead of a,b,c)

6. PLOS authors have the option to publish the peer review history of their article (what does this mean?). If published, this will include your full peer review and any attached files.

Reviewer #1: No

Reviewer #2: No

---

## [Author Response · Author response to Decision Letter 0]

19 Aug 2020

Tomáš Želinský, Ph.D.

Academic Editor 

PLOS ONE

August 17, 2020

Dear Dr. Želinský,

Thank you for reviewing our manuscript entitled "Bias in the Perceived Prevalence of Open Defecation: Evidence from Bihar, India" (manuscript #: PONE-D-20-16576). Your comments and those of the reviewers were highly insightful and enabled us to greatly improve the quality of our manuscript. We appreciate the suggested modifications and have revised the manuscript accordingly. 

In the remainder of this document, the reviewers’ comments are shown in bold, our response is shown in plain typeface, and sections from the revision are contained within quotation marks.

We hope that the revised manuscript and our accompanying responses will be sufficient to make our manuscript suitable for publication in PLOS ONE.

Thanks very much for your support, and we look forward to hearing from you soon.

Sincerely,

Jinyi Kuang

Reviewer #1: 

 1. How representative is the sample? This is important if the authors are trying to argue that their community level measure is a proxy for the community level norm. Can the authors compare characteristics against those in the census for example?

We thank the reviewer for raising this issue. Our sample is not designed to be representative at the state or district level. However, within the sampled districts we randomly selected Gram Panchayats and Town Panchayats (slums in municipal corporations were also selected randomly, but due to administrative constraints we do not claim representativeness in the district level for this type of geography). Within each selected cluster we randomly selected individuals from the list of all eligible individuals in each household within the cluster. Given the total sample size of 80 respondents per cluster on average, we infer that our sample is representative at the cluster level which are Gram Panchayats for rural areas and census wards for peri-urban areas and slums for urban areas. 

We see the value of using census level data to support this. However, our study was conducted in 2018, whereas the latest Census in India was conducted seven years ago, in 2011. We think that the sanitation scenario has changed considerably in Bihar since then because of subsequent central and state-level initiatives so a direct comparison is unfortunately not possible.

 2. Direct response questions asking about individual open defecation may suffer from socially desirable response bias while those about the community are less likely to suffer from such a bias. Do the authors think this is an issue? If so a caveat should be added in their results or conclusion.

This is a reasonable concern. To address the issue of social desirability bias, we asked a balanced question to gather information on individual-level open defecation behavior: “Some people defecate in the open; some people use a toilet, where did you defecate the last time you had to?”. We also collect observational data regarding toilet ownership and functionality and cross-checked behavioral measures. However, given the substantial sanitation improvement programs in our studied areas, we acknowledge that there might be potential under-reporting of individual-level open defecation (OD) practice. Accordingly, we added the following sentences to the limitation section:

 “The study site and India nationwide were subject to substantial sanitation promotion programs during the conduct of this study. Respondents may have under-reported open defecation practice due to social desirability bias. We tried to address this issue by asking balanced questions like “Some people defecate in the open; some people use a toilet, where did you defecate the last time you had to?” to reduce social desirability bias. We also collected observational data regarding toilet ownership and functionality, and cross-checked behavioral measures with these observational measures.”

 3. For their community level measure the authors use the questions "Some people defecate in the open; some people use a toilet, where did you defecate the last time you had to?" and then average across all respondents in the community. What is the distribution of these responses? Are the results robust to using the median? And importantly, are the results robust to using the community average of the question “In the past week, how often have you used a latrine to defecate? Never, occasionally, frequently, or every time?"” (as a community level measure). The reason I ask is that I am a little concerned with the time frame chosen by the authors for their community measure and the individual perceived prevalence. It is highly possible that people use both a toilet and open defecate and this intensity would not be accounted for with their current measures. To test if it matters, one can compare my suggested bias (community open defecation in the past week with perceptions about community defecation the last time) with the currently used bias.

The response to the question “Some people defecate in the open; some people use a toilet, where did you defecate the last time you had to?” are binary, with 1 = defecate in the open and 0=use a toilet. We therefore reported the count and percentage of each option in table 1. We think using the mean of this binary measure (number of people who OD over the total number of sampled respondents in a community) is more appropriate to reflect the community-level defecation rates than median or distribution because median is not well defined for this data type. 

Reviewer suggested aggregating individual’s defecation practice one week prior to the survey to construct the community-level OD rates, and compare it to individuals’ perception of OD prevalence regarding the last defecation event. We have several concerns using this approach: 1) To calculate the bias in the perceived OD prevalence, we would subtract the actual OD prevalence in the past week from the perceived OD prevalence of the past defecation event. Since these two measures have different timeframes, the constructed variable (i.e., bias) might not be easily interpreted; 2) an individual’s perception regarding the OD prevalence could also change depending on the time frame of choice. Reviewer concerned that using the last defecation event might underestimate the community-level OD rate than using the one-week time frame, yet the individual could also perceive a higher OD prevalence while using the one-week timeframe than the last defecation event. Given our measure of bias in this perception is a relative measure (subtract the actual OD prevalence in the past week from the perceived OD prevalence of the past defecation event), this concern should be resolved as long as these two measures are using the same timeframe.

 4. The authors may want to cite the recent paper by Leonardo Bursztyn Alessandra L. Gonzalez and David Yanagizawa-Drott. Misperceived Social Norms: Female Labor Force Participation in Saudi Arabia (accepted at American Economic Review) which focuses on differences between individual beliefs and perceptions about the community.

Thanks for the suggestion. We added this citation to our introduction section. 

 5. In the introduction the authors only specifically mention false consensus bias in the discussion of the results. Since this paper is essentially about this bias, it may be useful to mention this theory earlier in the introduction.

We added the following clarification to the introduction of the revised manuscript: 

“Specifically, people tend to show a false consensus bias, that is, perceiving their own behavior is more common than it actually is.”

 6. I assume the footnotes for Table 1 are incorrect please check (there are two references to a footnote a and no reference to a footnote c). On this note, I am a little confused about the differences between Bias in the perceived prevalence of open defecation and the Accuracy of the prevalence perception of open defecation (Table 1). The authors need to discuss the differences between these two definitions and results. The authors may want to consider using different labels. Further, according to Table 1, the bias measure is zero suggesting (as the authors state on line 133 of page 7) individuals have an accurate perception of open defecation prevalence. This is quite striking and not particularly discussed in the results. And why do the authors focus on the accuracy measure when discussing Table 1 (ignoring discussion of the bias measure) and then fail to use this accuracy measure in their estimation (Table 2, Figure 2) instead turning back to the bias measure. This seems to be inconsistent and not well explained.

We revised the notations in Table 1 to match the one used in the footnotes.

We also revised how bias and accuracy were presented in Table 1 to clarify the difference between accuracy and bias in the prevalence perception. Specifically, we presented the count and percentage of respondents who overestimate, underestimate and hold an accurate perception of OD prevalence. 

We added additional explanations regarding the accuracy and bias measures in the results section:

“We found respondents show an overall inaccurate perception of OD prevalence in their communities (Table 1). Specifically, respondents’ perception of OD prevalence deviated from the actual prevalence of OD by 17% on average (SD=0.15). Regarding the direction of this misperception, we found 45.3%% overestimate the OD prevalence and 51.5% underestimate the OD prevalence.”

 7. The authors should discuss results in Table 2 as they are important. At the moment the authors simply state “The analysis results were shown in Table 2.”

We added additional explanations of the analysis results shown in Table 2.

“In the regression analysis adjusting for the respondents’ gender, age, educational attainment, socio-religion, and socio-economic status, we found that compared to those who reported defecating in the open in the week prior to survey administration, those who use a toilet inconsistently (b= -0.10, 95% CI: -0.13, -0.08, p<0.001) and those who use a toilet constantly (b= -0.21, 95% CI: -0.23, -0.19, p<0.001) have significantly lower estimation of open defecation prevalence than it actually is (Table 2). The predicted value showed that those who defecated in the open in the week prior to survey administration perceived that open defecation was 13% more prevalent than the actual prevalence within their communities. In contrast, those who reported consistent use of a toilet perceived that open defecation was 8% less prevalent than the actual prevalence in their communities (Fig. 2).”

 8. Can the authors include the mean for “actual community open defecation prevalence” in Table 1.

We added the mean, as suggested to “Actual community-level open defecation prevalence” to Table 1.

 9. Where do Figure 2 results come from? Results appear to be different from those in Table 2

Table 2 presented coefficients and confidence intervals of the linear regression model. Figure 2 depicted the predicted values and confidence intervals of the bias in the perceived prevalence of open defecation by individual’s defecation patterns in the last week. The predicted values and standard errors were presented in the caption of Figure 2. 

Reviewer #2

 1. One minor comment: Superscripts in Table 1 referring to the Table notes are wrong (a,a,b instead of a,b,c)

Thank you for your comment. This has been revised.

---

## [Editor Report · Decision Letter 1]

21 Aug 2020

Bias in the Perceived Prevalence of Open Defecation: Evidence from Bihar, India

PONE-D-20-16576R1

Dear Dr. Kuang,

We’re pleased to inform you that your manuscript has been judged scientifically suitable for publication and will be formally accepted for publication once it meets all outstanding technical requirements.

Kind regards,

Tomáš Želinský, Ph.D.

Academic Editor

PLOS ONE

Additional Editor Comments (optional):

Thank you very much for addressing all concerns raised by the reviewers and for providing detailed descriptions and explanations of your changes. I am more than happy to recommend this paper for publication. 
---

## [Editor Report · Acceptance letter]

2 Sep 2020

PONE-D-20-16576R1 

Bias in the Perceived Prevalence of Open Defecation: Evidence from Bihar, India 

Dear Dr. Kuang:

I'm pleased to inform you that your manuscript has been deemed suitable for publication in PLOS ONE. Congratulations! Your manuscript is now with our production department. 

Kind regards, 

on behalf of

Dr. Tomáš Želinský 

Academic Editor

PLOS ONE